# Affective and Substance Abuse Disorders Following Abortion by Pregnancy Intention in the United States: A Longitudinal Cohort Study

**DOI:** 10.3390/medicina55110741

**Published:** 2019-11-15

**Authors:** Donald Paul Sullins

**Affiliations:** Department of Sociology, Catholic University of America, 620 Michigan Ave NE, Washington, DC 20064, USA; sullins@cua.edu

**Keywords:** abortion, pregnancy intention, depression, suicidality, substance abuse, panel data

## Abstract

*Background and Objectives*: Psychological outcomes following termination of wanted pregnancies have not previously been studied. Does excluding such abortions affect estimates of psychological distress following abortion? To address this question this study examines long-term psychological outcomes by pregnancy intention (wanted or unwanted) following induced abortion relative to childbirth in the United States. *Materials and Methods*: Panel data on a nationally-representative cohort of 3935 ever-pregnant women assessed at mean age of 15, 22, and 28 years were examined from the National Longitudinal Survey of Adolescent to Adult Health (Add Health). Relative risk (RR) and incident rate ratios (IRR) for time-dynamic mental health outcomes, conditioned by pregnancy intention and abortion exposure, were estimated from population-averaged longitudinal logistic and Poisson regression models, with extensive adjustment for sociodemographic differences, pregnancy and mental health history, and other confounding factors. Outcomes were assessed using the Diagnostic and Statistical Manual, Version 4, American Psychiatric Association (DSM-IV) diagnostic criteria or another validated index for suicidal ideation, depression, and anxiety (affective problems); drug abuse, opioid abuse, alcohol abuse, and cannabis abuse (substance abuse problems); and summary total disorders. *Results*: Women who terminated one or more wanted pregnancies experienced a 43% higher risk of affective problems (RR 1.69, 95% CI 1.3–2.2) relative to childbirth, compared to women terminating only unwanted pregnancies (RR 1.18, 95% CI 1.0–1.4). Risks of depression (RR 2.22, 95% CI 1.3–3.8) and suicidality (RR 3.44 95% CI 1.5–7.7) were especially elevated with wanted pregnancy abortion. Relative risk of substance abuse disorders with any abortion was high, at about 2.0, but unaffected by pregnancy intention. Excluding wanted pregnancies artifactually reduced estimates of affective disorders by 72% from unity, substance abuse disorders by 11% from unity, and total disorders by 21% from unity. *Conclusions*: Excluding wanted pregnancies moderately understates overall risk and strongly understates affective risk of mental health difficulties for women following abortion. Compared to corresponding births, abortions of wanted pregnancies are associated with a greater risk of negative psychological affect, particularly depression and suicide ideation, but not greater risk of substance abuse, than are abortions of unwanted pregnancies. Clinical, research, and policy implications are discussed briefly.

## 1. Introduction

### 1.1. Incidence and Significance of Wanted Pregnancy Abortions

Each year almost seven million induced abortions terminate pregnancies which the patient later assesses to have been planned or wanted. This estimate interpolates the rate of wanted pregnancy abortions found in this study (14.7%) to the estimated 56 million global abortions per year 2010–2014 reported by World Health Organization surveillance [1,2], reduced by 15% to ensure a conservative estimate. The resulting number is 6,997,200 annual abortions of wanted pregnancies. Although such wanted pregnancy abortions (hereafter “WPA”) are thought to be more psychologically distressing [3,4], long-term psychological outcomes following such abortions have not previously been studied.

The proportion of women obtaining WPAs is small compared to abortions of unwanted pregnancies (hereafter “UPA”) [5,6], but it is not trivial. Finer reports that 8% of pregnancies ending in abortion were reported as intended on the 2002 National Survey of Family Growth (NSFG) [7]. In the U.S. cohort examined for the present study, 14.7% of total abortions, involving 18.3% (95% CI 16–21) of ever-aborting women, were of pregnancies reported as wanted. Data on the reasons for such abortions are not available. The Task Force on Mental Health and Abortion of the American Psychological Association (APA) has speculated that “[w]omen who terminate wanted pregnancies typically do so because of fetal anomalies or risks to their own health” [5]. No doubt this accounts for a portion of such terminations, however, life-threatening fetal anomalies occur at only about one-tenth the rate of WPA. In the Add Health data, 51 of every thousand pregnancies reported by women ended in a wanted pregnancy abortion, but major cardiac deformity, the most common neonatal abnormality, only occurs in about four of 1000 births, and the most common fetal screening procedure, an ultrasound at 18–23 weeks, is not more than 60% successful in identifying such abnormalities [8]. Even by the most generous estimate, and counterfactually assuming no abortion concealment, no more than a fifth of the reported wanted child abortions could be due to fetal abnormalities. Although more frequent debilitating anomalies such as Down Syndrome or Trisomy 13 are also increasingly indications for abortion. Whatever the rate, the U.S. National Academy of Science’s 2018 Report on abortion care advises: “Terminations of pregnancies due to fetal abnormalities may have very different psychological consequences than abortions for unwanted pregnancies” [9].

Recent research has supported the APA Task Force’s speculation that abortions of wanted pregnancies may be “associated with more negative psychological reactions among women who have had an abortion” [4]. Intake studies of women seeking an abortion have found that feeling pressured by others and low confidence in the abortion decision increased the risk of poor post-abortion coping [10]. Rocca et al. found that women aborting planned pregnancies had lower odds of reporting that the abortion was the right decision and had more negative emotional reactions immediately following the abortion [11].

Retrospectively assessed wantedness may manifest at the time of the procedure as ambivalence or being conflicted with the intentions of others. In abortion clinic intake surveys 14–24% of patients indicated that she was acceding to the wishes of her partner, 6–8% to the wishes of her parents, and 8–12% reported health problems as one reason among others for the abortion [10]. More detailed studies of decision-making among women seeking abortions have found that 10% of minor women and 3% of adult women indicated that they sought an abortion primarily because of pressure from others [12], 8% scored high on a measure of “decisional conflict”, and 5% indicated that they “preferred to have the baby” [13]. Research in the National Survey of Family Growth (NSFG) also suggests that discordant partner intentions may influence the abortion of a first pregnancy otherwise intended by the woman [14].

### 1.2. Literature Review: The Abortion and Mental Health Controversy

The literature on abortion and mental health is sharply divided between studies finding psychological risks following abortion to be significant and persistent [15,16,17,18,19,20,21,22] and those finding them to be negligibly small and transient [23,24,25,26,27,28]. Some studies have found a positive association between having an abortion and a range of difficulties [17,19,29,30] including suicidality [18,19], depression [15,17,19,31,32,33], anxiety [15,34], and substance abuse [19,35]. Other studies have reported weak or null results on all these outcomes [23,24,25,27,36,37]. Competing reviews have vigorously critiqued the methodology of those on the other side of the divide [6,28,29], (for reviews of the entire controversy see [30,38,39,40,41]). The controversy has prompted successive methodological improvements, including insistence on population representative data, comparison samples, validated measures of mental disorder, substantial controls for prior medical history, confounding covariates and alternate pregnancy outcomes, and a preference for longitudinal cohort data with long term (at least 7 years) follow up [25,42].

Recent research has focused on overcoming the limitations of retrospective cross-sectional fertility data in favor of rigorous longitudinal designs which can more clearly establish the time order of cause and effect [4,18]. Most studies using longitudinal data, however, have only analyzed reported outcomes at terminus, which does not make full use of the potential of the data to model temporal sequence [17,26,43,44,45]. Such studies have also been limited by short follow-up periods following abortion, from as little as one month [46] to no more than five years [23,24,25,31,46,47]. Some of the strongest evidence to date has come from three longitudinal studies by Fergusson, Pedersen, and Sullins of women in New Zealand, Norway, and the United States that measured health outcomes and abortion status over at least three points in time from adolescence into the late 20s. All three studies found significant post-abortion increases in the risk of affective and addictive disorders, including depression, thoughts of suicide, anxiety, and abuse of illicit drugs, marijuana, or alcohol [20,35,48,49].

A series of recent studies based on clinical samples of women not permitted to finalize a desired abortion for medical or legal reasons, known as the Turnaway Study [23,25], has claimed to provide strong evidence of minimal mental health disorders [9]. However, this study design cannot examine the crucial difference in outcomes between women who choose to terminate a pregnancy by abortion and those who do not. The Discussion section addresses this issue further.

### 1.3. Neglecting Wanted Pregnancy Abortions

Reviews by American and British medical associations have asserted that abortion is not psychologically harmful, but only with respect to unwanted or unplanned pregnancies. The APA Task Force found, for example, that “among adult women who have an *unplanned pregnancy* the relative risk of mental health problems is no greater if they have a single elective first-trimester abortion than if they deliver that pregnancy (emphasis in original)” [4]. The U.S. National Academy of Sciences (NAS) reached a similar conclusion when examining “whether women who have an abortion experience more mental health problems than women who deliver an *unwanted pregnancy* (emphasis added)” [9], as did Great Britain’s Academy of Medical Royal Colleges (AMRC) on the question posed as follows: “Are mental health problems more common in women who have an induced abortion, when compared with women who deliver an *unwanted pregnancy* (emphasis added)” [6]?

### 1.4. Study Aim and Hypothesis

Despite such claims, no study has yet examined whether long-term mental health problems for women following UPA differ from those following WPA. The present study aims to amend this defect in the literature. The question has important implications for policy since, if the APA is correct that WPAs are more psychologically harmful, neglect of them may lead to the systematic understatement of mental health problems following abortion. The hypothesis of this study is that, compared to the corresponding births, WPAs will be associated with greater mental health risk than UPAs. Designed to address all of the above-noted methodological concerns, the present study examines the risk of seven clinical disorders, defined by Diagnostic and Statistical Manual, Version 4, American Psychiatric Association (DSM-IV) or other well-validated criteria, for women exposed to WPA compared to those exposed to UPA, relative to corresponding births, in a nationally representative cohort of ever-pregnant U.S. women assessed at three points over thirteen years.

## 2. Materials and Methods

### 2.1. Data

Initiated by a consortium of 18 federal agencies, the National Longitudinal Study of Adolescent to Adult Health (Add Health) was designed to be the most extensive study of the health-related behaviors of U.S. adolescents during the transition to adulthood. In 1995, researchers obtained extensive measures of behavior, attitudes, and well-being from interviews with a nationally representative sample of 20,745 US adolescents (Wave I) selected from a school-based multistage cluster sampling frame stratified by school size and type, urbanicity, ethnicity and region [50]. After a one-year follow-up at Wave II, which is not used in this analysis, 12,288 members of the original sample, representing over 80% of those available, completed follow-up interviews at both Wave III in 2001–2002 and Wave IV in 2008–2009. The resulting data provide representative longitudinal health measures for this national cohort at mean ages of 15.1 years (SD 1.74, range 11–21) at Wave I (baseline), 22.0 years (SD 1.77, range 18–28) at Wave III and 28.5 years (SD 1.79, range 24–34) at Wave IV (terminus). A full description of the Add Health sample design is available at https://www.cpc.unc.edu/projects/addhealth/design/.

Lifetime pregnancies and outcomes reported at all three Waves were combined to obtain a comprehensive pregnancy history. By Wave IV, 6139 ever-pregnant female respondents had reported 14,472 unique pregnancies at one or more Waves, consisting of 8900 pregnancies ending in birth, 2015 ending in induced abortion, and 2077 ending in miscarriage or other involuntary pregnancy loss. Excluding pregnancies which were ongoing at interview or for which a clear outcome was not reported left 5579 women with completed and clearly defined pregnancies at Wave IV (Wave IV Only Sample); 4523 of these reported information at all three Waves (Full Sample). Non-response on one or more of the 25 covariates and demographic control variables in the analysis reduced the sample by an additional 588 cases (13.0%), resulting in an analytic sample of 3935 cases with information on all variables at all Waves (Analysis Sample). The longitudinal analysis models employed information from all three Waves, estimating from a maximum of 11,805 wave-by-respondent cases.

Table 1 reports descriptive statistics for the variables in the analysis, comparing the Wave IV Only Sample, the Full Sample, and the Analysis Sample. Due to incompatible weights the table reports unweighted data, which overstates the population differences. Nonetheless, all differences are small, and most are not significant. Consistent with sample mortality, the Wave IV Only Sample, compared to the Full Sample, had slightly higher poverty, lower income, and education, and were more likely to have experienced parental abuse or forced sex. Anomalously, they were also slightly less likely to report drug or alcohol abuse. Compared to the Full Sample, those in the Analysis Sample were slightly younger and less likely to be white or to report an abortion, parental sex abuse, forced sex or being depressed. They also had slightly higher educational attainment but lower income at Wave IV. These differences are consistent with item non-response. Sensitivity analyses have also found negligible non-response bias for health risk measures at Waves III and IV of Add Health [51,52]. As noted, the present study utilizes population weights designed to adjust for cross-Wave sample differences, which assures that the Wave III as well as the Wave IV sample “adequately represents the same population surveyed at Wave I” [52].

The Add Health data are publicly available in both a restricted version, used in this study, and an unrestricted version with reduced cases. As a secondary analysis of existing public data, the Catholic University of America Institutional Review Board certified this study (Sponsored Research Project 200223) as exempt from data collection protocols under 45 CFR 46.101, and granted ethical approval for the data use protocol (Protocol 14-052), on 15 September, 2017. Information on how to obtain the Add Health data files used in this study is available at http://www.cpc.unc.edu/addhealth. 

### 2.2. Measures

The present study examined seven mental health and substance abuse outcomes, imposing five demographic controls and 20 covariates. The variables are listed in Table 1. The description of most covariates has been previously reported in a prior study [49]. The account that follows describes only those variables unique to the present study.

To assess psychological outcomes, the present study included opioid abuse in addition to the previously-described outcomes of depression, anxiety disorder, suicidal ideation, illicit drug abuse, cannabis abuse and alcohol abuse. Opioid abuse was measured at Wave I by a general question that included using “pills…without a doctor’s prescription”, and at Waves III and IV by more detailed questions that asked about the use of pain killers or opioids, naming four to six common brands such as Vicodin or Percodan, either without a prescription or “in larger amounts than prescribed, more often than prescribed, for longer periods than prescribed, or that you took only for the feeling or experience they caused?”

Measures of pregnancy outcomes and intentions were compiled from retrospective accounts at each Wave. Pregnancies ending in miscarriage, stillbirth, ectopic pregnancy, or other pregnancy loss were combined into a single category of “involuntary pregnancy loss”, resulting in pregnancy outcome categories of birth, abortion, and involuntary pregnancy loss. For each pregnancy, respondents were asked to respond yes or no to the question: “Thinking back to the time just before this pregnancy with {initials}, did you want to have a child then?” Women responding “yes” for any aborted pregnancy were coded as having experienced WPA, yielding analysis categories of never abortion, UPA only, and ever WPA. The corresponding procedure for births yielded categories of never birth, wanted pregnancy births (WPB) only, and ever unwanted pregnancy birth (UPB). A woman could thus be in both birth and abortion categories for a given comparison, which accurately reflects the complex interaction of abortions and births in women’s pregnancy decisions over time.

To better isolate the effect of abortion, the analysis adjusted for multiple covariates that were significantly associated with prior or current mental distress or the precipitation of unwanted pregnancy. To fully adjust for prior mental health history, all seven outcomes were entered as continuous measures at Wave I, and lagged values of all outcomes at any prior waves, that is, at Wave I for Wave III measures, and at Waves I and III for Wave IV measures, were entered as covariates at every wave.

Other covariates fitted included retrospective measures of childhood family conditions, including poverty status, parental education, and any physical, sexual or verbal abuse; conditions measured at baseline, including neuroticism, conduct problems, community integration, and school grade point average; conditions measured at terminus, including educational attainment, current relationship satisfaction, and lifetime rape victimization; and time-dynamic covariates measured at all three Waves, including income, marital status, and intimate partner violence (IPV) victimization. Detailed descriptions of these measures have been previously published [49].

### 2.3. Study Design

The unit of analysis was not simply pregnancy intention or outcome, but women who had experienced the pregnancy outcome-by-intention combinations of interest. Accordingly, the analysis examined four design conditions: (1) ever-aborting women who had aborted only unwanted pregnancies (unwanted pregnancy abortion (UPA); Wave IV *n* = 807); (2) women ever giving birth who had brought to term one or more unwanted pregnancies (unwanted pregnancy birth (UPB); Wave IV *n* = 1913); (3) women ever giving birth who had brought only wanted pregnancies to term (wanted pregnancy birth (WPB); Wave IV *n* = 1345); and (4) ever-aborting women who had aborted one or more wanted pregnancies (wanted pregnancy abortion (WPA); Wave IV *n* = 210). The conditions were not mutually exclusive. Conditions 1 and 2 were compared to determine the effect of abortion relative to birth for unwanted pregnancies; conditions 3 and 4 were compared to determine the effect of abortion relative to birth for wanted pregnancies.

Relative risk ratios (RRs) for each association of abortion by pregnancy intention with mental disorder were computed using population-averaged longitudinal logistic regression models. Incidence rate ratios (IRRs) for the number of affective disorders, substance abuse disorders and total number of mental disorders were estimated from the corresponding poisson models. The RRs and IRRs are interpreted as the ratio of the probability of experiencing the indicated psychological outcome (for example, depression) conditional on being in each state of the independent variable (for example, WPA), averaged (or pooled) over all time periods. Also computed were the corresponding odds ratios (OR), that is, the ratio of the odds of experiencing the indicated psychological outcome (for example, depression) for those in the indicated state (for example, WPA) compared to the odds of experiencing the outcome for those not in the indicated state. Model fit was assessed using the Archer–Hosmer–Lemeshow F-adjusted mean residual test [53]. Analyses were performed with Stata 13 statistical software, incorporating the design features of the survey following published guidelines [54]. RRs computed from random effects and fixed effects models were very similar (not shown). 

## 3. Results

Table 2, Table 3 and Table 4 presents the unadjusted prevalence for the seven independent and three summary measures of psychological disorder which comprise the outcomes examined in this analysis. These tables are presented for preliminary context and illustration only. The chi-square test of equal means for the summary measures shown in Table 2, Table 3 and Table 4 employed James’ adjustment for possible sample heterogeneity [55]. Dispositive analyses which employ parametric models and impose covariate constraints are presented below.

Several trends in the prevalence tables are worth noting. First, at all three Waves, summary mental health problems for women experiencing abortion were consistently higher than for those not experiencing abortion, including those who had not yet been pregnant. Second, at all three waves, every summary measure of mental health problem prevalence was significantly affected by the women’s experience with abortion. The number of individual outcomes affected by abortion increased over the three Waves. At Wave I, just three of the seven individual outcomes were significantly affected by abortion experience; by Wave II, five were; and by Wave IV, six of the seven were so affected. Alcohol abuse and cannabis abuse were affected at every Wave, but the effect of abortion on anxiety was not significant at any Wave. Third, the general trend for mental health problems is shaped like an upside-down “U” across the three waves, or temporally by age: problem prevalence generally increased from Wave I to Wave III (average ages 15 to 22), then declined by Wave IV (average age 28). For all women in the sample, the summary count of all seven mental health problems was 1.05 at Wave I, rose to 1.40 at Wave III, then declined to 1.01 at Wave IV. This trend is the resultant of two opposing trends by type of disorder: summary substance abuse disorders more than doubled, from 0.44 to 0.99, between Waves I and III, then dropped sharply, to 0.56, by Wave IV; by contrast, summary affective disorders declined, from 0.62 to 0.40, between Waves I and III, then rose slightly, to 0.45, by Wave IV. These opposing temporal trends were replicated in each category of pregnancy and abortion experience. Whether these observed differences are statistically significant or persist in the presence of covariate adjustments are questions to be examined below. 

Table 5 reports adjusted longitudinal regression models testing the effect of intention on the question whether the abortion of a pregnancy is associated with greater subsequent mental health disorders compared to bringing the pregnancy to birth. The table shows the relationships between abortion history and subsequent mental health for ever-aborting women, measured by longitudinal regressions utilizing measures from all three waves of data and that adjust for all other pregnancy outcomes, covariates, and confounders identified in this analysis. Abortion and birth history are characterized by dichotomous measures representing whether a woman by the given age had ever experienced the abortion of a wanted pregnancy compared to only the birth of a wanted pregnancy, or only the abortion of an unwanted pregnancy compared to the birth of an unwanted pregnancy.

(2) Women ever exposed to WPA experienced an increased risk of affective mental health disorder compared to women exposed only to UPA, with RRs ranging from 1.61 to 1.77. This trend is summarized in the fact that women from age 15 to 28 (on average) who ever experienced WPA, relative to women who brought only wanted pregnancies to term, experienced overall rates of affective disorders 1.43 (95% CI 1.08–1.89) times higher (*p* < 0.05) than those exposed only to UPA.

(3) Women ever exposed to WPA experienced a reduced risk of substance abuse disorder compared to women exposed only to UPA, with RRs ranging from 0.75 to 0.99. However, the confidence intervals for these RRs all included unity, and overall rates of substance abuse disorders were no different, at 0.99 RR (95% CI 0.75–1.31), for women exposed to WPA compared to those exposed only to UPA.

These findings support the hypothesis that women’s estimated risk of mental health disorder with abortion relative to birth is lower for unwanted pregnancies only than it is for all pregnancies. However, the corresponding claim that women exposed to abortion relative to birth would experience higher risk of mental health disorders with WPA compared to UPA was supported for affective disorders—for which the risk for women exposed to WPA was higher, at 1.69 (95% CI 1.31–2.18), than with only UPA, at 1.18 (95% CI 1.00–1.40)—but was not supported for substance abuse disorders—for which the risk for women exposed to WPA was not different, at 1.99 (95% CI 1.53–2.58) than with only UPA, at 2.01 (95% CI 1.69–2.38).

The risk ratios (RR) shown in Table 5 depend not only on abortion but also childbirth, expressing the ratio of the probability of mental health problems with abortion and with childbirth. To examine these effects separately, Table 6 expresses the corresponding odds ratios (OR) for each pregnancy outcome independently of the other. The ORs can be interpreted as expressing the direct or unique effect of abortion exposure, as distinct from its effect relative to birth. Appendix A
Table A1 compares directly these similar but unique measures.

Consistent with Table 5, in Table 6 women exposed to abortion experienced reduced risk of affective problems, but increased risk of substance abuse problems, if they had experienced only UPA rather than ever WPA. On the other hand, women experienced increased risk of both affective and substance abuse problems when exposed to childbirth of only unwanted pregnancies compared to birth of one or more wanted pregnancies.

## 4. Discussion

### 4.1. Main Findings

The most notable finding of this study is that WPA was associated with greater risk of affective disorders but was not associated with greater risk of substance abuse disorders compared to only UPA. Substance abuse disorders, while strongly associated with pregnancy outcome, at about twice the rate with abortion than with childbirth, do not appear to be related to pregnancy intention. In the present findings (Table 5), excluding WPA when estimating distress associated with abortion substantially understated the risk of affective disorders, by 72% from unity (1.31/1.18), and total disorders, by 21% from unity (1.74/1.61). On the other hand, excluding WPA modestly understated overall rates of substance abuse disorders, by 11% from unity (2.12/2.01).

The substantially increased rate of affective problems with WPA was largely attributable to notably high comorbid risks of depression (2.22) and suicide ideation (3.44) relative to childbirth. Elevated depression risk is consistent with prior studies finding that “women who abort due to a poor prenatal diagnosis are at higher risk of post-traumatic stress and depression than women who continue with pregnancy” [56,57]. Indeed, the most persistent finding of research showing psychological risk with abortion is an association of abortion with subsequent indicators of depression [19,20,29,30,31]. Long-standing research has also confirmed a strong link between abortion and suicidality. Luo et al. found that the risk of suicidal ideation was almost twice as high (OR 1.89, *p* < 0.05) among young single Chinese women who had had an induced abortion compared to those who had not, after controlling for co-occuring morbidities including depression, anxiety, and low self-esteem [22]. Gissler, reporting on linked Finnish health records, found that suicide risk doubled for Finnish women with a recent induced abortion [58]. Subsequent study of the same records further found that teenage abortion, compared to childbirth, predicted premature death from all causes among Finnish women in their 20s [59].

Barber et al. proposed two mechanisms that may lead to depression in unwanted pregnancy childbearing that may also apply to WPA. First, they suggested that unwanted childbearing may be associated with a context of relational discontinuity including marital conflict and lack of social support which have been associated with depression in general [60]. Such conditions are also likely to be associated with wanted abortions. Second, they proposed that unwanted childbearing “can be considered an uncontrolled and undesired event: the mother did not want to have another child but she did anyway” [60]. Analogously, with WPA the mother did want to have a child but had an abortion, nonetheless. Examining population data, Cha and colleagues found that “[conceiving couples] with discordant pregnancy intentions were significantly more likely to have induced abortion, even in women who desired pregnancy” [14]. Such a contradiction, Barber et al. suggest, “is likely to lead to feelings of powerlessness. … [which are] particularly strong determinants of depression and anxiety” [60]; see also [61,62,63]. Confirming this idea, Kimport et al.’s analysis of open-ended descriptions from women emotionally distressed by abortion found that “negative outcomes were experienced when the woman did not feel that the abortion was primarily her decision (e.g., because her partner abdicated responsibility for the pregnancy, leaving her feeling as though she had no other choice) …” [64]. In a sample of post-abortive Finnish women, Kero et al. found that of the 21% who were experiencing “severe emotional distress”, a quarter (25%) “clearly stated that they wanted to give birth”, and another 42% were ambivalent [65].

### 4.2. Understating Post-Abortion Psychological Distress

Notwithstanding the null findings for substance abuse in this study, the restriction of research attention to UPA has nonetheless contributed to widespread understatement of post-abortion psychological distress. The reason for this is that affective outcomes have been the concern of the overwhelming majority of studies, and the almost exclusive focus of official reviews, of abortion and mental health. In the most recent official review by the NAS, after listing the major studies captured in its search of the literature, the authors note: “Most of the studies focused on whether abortion increases women’s risk of depression, anxiety, and/or posttraumatic stress disorder (PTSD)” [9]. Only three studies in the review examined substance abuse measures at all, [17,35,66] and only one study did so exclusively [35]. The NAS’s conclusion denying distress associated with abortion relied heavily on the findings of a single study project, the Turnaway Study, which measured mental health primarily by questions on depression, anxiety and suicidal ideation [9]—the same three affective disorders which were found in the present study to be significantly understated when considering only UPA.

The Turnaway Study illustrates both the importance and the conceptual difficulty of considering pregnancy intention when assessing post-abortion mental health. The project followed groups of women granted or denied clinic abortions for 5 years, with results reported in several related studies [23,25]. Pregnancy intention was ignored when assessing mental health levels at the 5 year follow-up, reported by Biggs et al. [25], despite the fact that women in the sample with planned pregnancies were significantly more likely to report negative emotions (regret, sadness, anger, guilt) at 3 years post-abortion [66]. Ignoring this evidence, which was contrary to their hypothesis, Biggs et al. went so far as to criticize the possible use of wanted pregnancy comparison groups by other researchers since, they claimed, “women with wanted pregnancies may differ on several confounding factors from women seeking abortions” [25]. The construction of this sentence, which contrasts “women with wanted pregnancies” with “women seeking abortions”, illustrates the common error which assumes that all abortions must involve unwanted pregnancy. The AMRC review also reports that it “assumed that all abortions were due to unwanted/unplanned pregnancies unless explicitly stated otherwise” [6]. This subtle bias is understandable on the part of those who provide or support a policy of minimally restricted abortion access, since the mirror image of the goal that births would only result from wanted pregnancies is the expectation that abortion would only result from unwanted pregnancies. Almost twenty years ago the U.S. Centers for Disease Control (CDC) empaneled a Working Group on Unintended Pregnancy to address this error, which cautioned: “Measures usually consider all abortions to be the result of unintended pregnancies. However, … decisions about abortion are driven not only by pregnancy wantedness, but also by the extent to which a woman accepts or rejects abortion as a way of resolving an unwanted pregnancy” [67].

### 4.3. Clinical Event or Life Event?

The fact that pregnancy intention is strongly associated with mental health outcomes following abortion suggests that adjustments related to the abortion procedure itself are not the only factors, and may not be the major factors, involved in women’s psychological reactions to abortion. Contrary to research claiming that unwanted pregnancy childbearing increases women’s risk of mental health difficulties [5,6,68], in the Add Health data examined in the present study women who gave birth to unwanted pregnancies consistently experienced lower risk of negative mental health compared to those who had an abortion (see Table 6). For all pregnancies, the absolute risk of mental health problems was elevated 21% with abortion (see Appendix A
Table A1) but was reduced 30% with childbirth (not shown). Although the absolute IRR of summary mental health problems with WPA was reduced by 8% compared to UPA, the IRR for the corresponding comparison with childbirth was reduced almost twice as much, by 15% (see Table 6). For all three aggregate outcomes shown in Table 6, pregnancy wantedness significantly reduced unique distress with childbirth (evidenced by *p*-values of 0.000 to 0.016) but did not significantly affect unique distress with abortion (evidence by *p*-values of 0.100 to 0.427). The psychological cost of an abortion thus reflects not only exposure to the risk of the procedure and subsequent coping, but even more the opportunity costs of intervening in the life course to deny or defer childbirth. Extending the comparison, it is quite possible that the clinical risk of an abortion procedure could be negligible while the life event risk is quite large.

The mixed findings in the literature on abortion and mental health may thus reflect, in part, differing study designs that model the experience of abortion either more narrowly, as a clinical event, or more broadly, as a life event. How broadly or narrowly the experience of abortion should be modelled in relation to particular policy or social questions may be a matter of debate and legitimate diversity among scholars, with no one answer applying to all women or all sociopolitical contexts. A concern only with direct clinical outcomes may be addressed by a more constrained or narrow model, whereas a concern for all possible psychological distress associated with an abortion would indicate the use of a broader life event model. At minimum, a life event model should include subsequent birth and relational history, and, since abortion is an elective procedure conditional on pregnancy, predictors and consequences of the index pregnancy and the decision to have an abortion as well as the clinical experience itself.

On the range I am describing, the Turnaway Study, which narrowly models only women who have chosen to have an abortion, is an appropriate design for information about the clinical experience of an abortion, but a weak design for the life event. Ditzhuijzen et al. have recently made a similar critique, noting that “information about the consequences of the whole life event … is lost when focusing on the specific effect of abortion versus denial of abortion” [69]. Other recent studies have also acknowledged the importance of considering pregnancy intention when examining a possible link between abortion and mental health. Lappalahti and colleagues, in a study of mental health outcomes following early abortion in the Finnish registry data, noted the “intentionality of the pregnancy” as an unmeasured confounding factor that limited their results [70]. Mota et al. and Munk-Olsen et al. also both acknowledged that the limitations of their studies included a lack of information about whether the aborted pregnancy was wanted or unwanted [19,24].

Effective capture of life-event outcomes typically requires longer follow-up. Studies examining women at least a decade following abortion, as the present study has done, have all found increased risk of psychological disorders relative to women with no abortions [18,19,20,35,48,49]. The results of the present analysis regarding exposure to abortion overall are very similar to those of the aforementioned longitudinal studies by Pedersen [35] and by Fergusson and colleagues [48], which consistently found higher risk of psychological distress for women ever exposed to abortion. Fergusson, employing similar models and covariate adjustments, found a 1.37 RR (compared to 1.21 in the present study) for number of mental health problems, after examining comparable longitudinal data for a cohort of 500 New Zealand women from ages 15 to 30 [48]. Overall, these studies, as does the present study, support a middle position between claims that “abortion has large and devastating effects on the mental health of women” and claims that “it is without any mental health effects” [48].

### 4.4. Abortion Under-Reporting and Generalizability

A persistent difficulty for any study using data on abortions in the United States is the difficulty of accurately counting abortions and/or post-abortion distress. Typically only perhaps half of U.S. abortions are captured on retrospective fertility surveys [71,72,73]. The exact extent of under-reporting is unknown because non-compliance by local health departments on national surveillance surveys is also high. Add Health used a computer-assisted anonymous data collection method which is known to reduce under-reporting [50], however the problem of under-report bias is unavoidable for any study of abortion using self-reported data. To date, alternatives to retrospective self-reports to measure both abortions and emotional outcomes more accurately have not yielded improved results. National government funding for abortions is proscribed in the United States, which limits the scope of record linkage measures. Studies based on clinic surveys have avoided high abortion concealment, but at the expense of even higher sample mortality—as much as 82% to five-year follow-up in the Turnaway Study [11]—resulting in even lower generalizability. Such studies also systematically under-report abortions in other institutions, such as surgical abortions in hospitals or medical abortions in doctor’s offices.

Tierney has recently suggested that abortion capture on Add Health is unusually low, at only 35% [74], however her study analyzed only 1833 abortions reported at the Wave IV interview. The present study includes an additional 419 abortions reported at Waves I and III which were not reported at Wave IV, or which were excluded or uncounted by Tierney at Wave IV, increasing abortion capture by 23%. The present study also includes an additional 481 abortions reported at the Wave III interview that were not reported at Wave IV, increasing effective abortion capture at Wave III by 58%. Since time to recall has been shown to have a strong inverse effect on abortion capture rates (in Tierney’s analysis the estimated percent of abortions captured at the 2008 Wave IV interview rises from 20% for 1994 abortions to 51% for 2007 abortions) [74], the inclusion of abortions reported at the earlier Wave III interview may have a disproportionate effect on increasing the overall longitudinal abortion capture rate. Examinations of particular demographic groups have found no differential under-reporting of abortion on Add Health [75], a finding Tierney confirms for racial groups [75]. Moreover, the unit of analysis for the present study is not total abortions, but ever-aborting women, for which the capture rate is somewhat higher, since a portion of higher abortion multiples are reported as fewer abortions than as no abortions. In sum, the proportion of ever-aborting women captured on Add Health in the present study is at least in line with that of comparable fertility surveys and does not appear to be subject to evident bias. As is often the case with population data, the Add Health data used in this study are, despite some weaknesses, nevertheless among the best measures we have for the study of abortion, and have been the basis for numerous and recent prior studies of abortion [20,21,22,23,24,25].

Furthermore, because the effect of under-reporting bias on mental health measures can plausibly be specified, abortion concealment is a much smaller problem for studies of abortion and mental health than may be the case with other topics. Abortion non-disclosure, which is comparable to that for miscarriage or other negative pregnancy outcomes, is generally interpreted to reflect similar social desirability bias or, put negatively, perceived or internalized stigma [76,77,78], as well as indications of psychologically distressing guilt and grief reactions [79,80,81]. As Cowan has recently observed, “women who terminate pregnancies are much more likely to have feelings of guilt and shame after the procedure than women who miscarried” [78] which affects disclosure. Whatever the mechanism, there is no dispute that abortion misreporting operates in only one direction, that of concealment. To my knowledge, no analysis or clinical report has ever found or suggested that women over-report their abortions or pretend to have an abortion that they knew they did not have. Therefore, in a fertility survey based study of post-abortion distress such as the present study, although the reported levels of psychological distress may not be generalizable as population point estimates, they can reasonably be taken as the lower bound of post-abortion distress. This is not an insignificant inference for interpreting the findings, since it means that, under the premise of high rates of concealment, the true population levels of post-abortion trauma, which would remain unknown, are very probably higher than the levels reported. Nonetheless, Tierney’s measured conclusion that “survey self-reports of abortion need to be evaluated, contextualized and used with caution” [74] is an important caution to bear in mind for any study of abortion using U.S. data.

### 4.5. Limitations

These findings are subject to a number of other limitations. Unlike conditions prior to pregnancy, for conditions following pregnancy or abortion, such as suicidality, IPV, substance use disorder, and stress, the direction of causation cannot clearly be specified and is probably not monotonic. Selective mortality associated with depression or suicidality may also lead to an understatement of the relation between abortion and these effects. Some measures were retrospective, subject to recall bias and revision. In particular, the use of retrospective measures of pregnancy intentions may limit the applicability of these findings to settings or studies involving prospective reports of pregnancy intention. However, studying unintended births, which are formally similar, Joyce and colleagues found “no evidence that the retrospective assessment of pregnancy intention produces misleading estimates of either the number or consequences of unintended births” [82]. Although no non-experimental study design can rigorously establish causation, Major argues that a case for causality can be made from longitudinal studies that establish covariation between a prior abortion and subsequent mental health outcomes with sufficient controls for other possible causes [5]. The present study meets these conditions.

## 5. Conclusions

Research and policy estimates that consider only unwanted pregnancies risk substantially understating the long-term risk of affective disorders, particularly depression and suicidality following abortion, while overstating the risk of substance abuse disorders. Failure to distinguish outcomes by pregnancy intention, moreover, may mischaracterize overall mental health risk due to abortion by flattening intention-specific risks. Future research is needed to specify more clearly the association of intentions with level and type of distress in the relational context of abortion.

The relative distress of abortion versus childbirth is more a product of the stronger unique mental health benefits of childbirth and children than of the weaker unique mental health deficits of abortion. The experience of deciding upon, experiencing, and recovering from the termination of a pregnancy brings many life factors to bear for women, all of which may influence subsequent mental health. For these reasons, it may be more accurate to conceive of an abortion, not as a discrete cause of mental health outcomes (a clinical event), but as one factor in a complex of influences (a life event) that together affect a woman’s level of psychological well-being or distress.

The justification of abortion on grounds of mental health in American jurisprudence and medical care assumes that procuring an abortion will typically result in less psychological and life distress than will bringing the pregnancy to term. The present study adds to a strong research consensus that challenges the basis of that assumption. Although some studies have found little to no psychological distress associated with abortion, to date no study has documented mental health benefits for women from abortion.

To improve women’s health, it is recommended that therapeutic interventions for women experiencing distress from abortion determine pregnancy intention as part of the diagnostic or targeted care plan. In clinical intake, efforts to identify the complex and relational sources of coercion, indeterminacy and ambivalence in women’s experience of pregnancy and abortion, rather than compressing this complexity into binary oppositional categories of “unwanted” versus “wanted”, may also help practitioners to better understand, address and ameliorate the psychological stress women may be facing in the life experience of abortion.

## Figures and Tables

**Table 1 medicina-55-00741-t001:** Descriptive characteristics for Wave IV only, full, and analysis sample: ever-pregnant women, add health Wave IV (unweighted).

Variables in the Analysis	Wave IV Only Sample N = 5579	Full Sample N = 4523	Analysis Sample N = 3935	Difference W4-Full (*t*-test)	Difference Full-Analysis (*t*-test)
Age (mean)	28.59	28.56	28.54	0.06	0.02
Nonwhite (%)	38.83	38.55	37.74	0.13	0.01
Ever abortion (%)	23.28	23.30	22.64	0.96	0.01
Number of abortions (mean) if ever abortion	1.55	1.54	1.52	0.28	0.29
Ever wanted pregnancy abortion (WPA) if ever abortion (%)	18.17	18.58	18.29	0.44	0.56
Ever unwanted pregnancy abortion (UPA) if ever abortion (%)	88.14	87.94	88.10	0.65	0.69
Covariates measured at Wave 1					
Grade point average	2.81	2.81	2.82	0.85	0.00
Parent B.A. (%)	21.35	22.10	22.57	0.01	0.06
Family poverty (%)	15.45	14.57	14.28	0.00	0.18
Trouble in school (scale mean)	0.997	0.996	0.993	0.74	0.39
Neighborhood integration (scale mean)	2.21	2.21	2.21	0.22	0.58
Self-esteem (scale mean)	4.00	4.00	4.00	0.80	0.53
Covariates measured at Wave 3					
Parent physical abuse (%)	19.78	18.96	18.68	0.00	0.24
Parent sex abuse (%)	8.70	8.01	7.65	0.00	0.04
Parent verbal abuse (%)	52.63	51.77	51.23	0.01	0.08
Lower income (%)	62.03	61.61	61.55	0.02	0.82
Covariates measured at Wave 4					
B.A. (%)	24.25	25.43	26.61	0.00	0.00
Relationship Satisfaction (scale mean)	0.885	0.881	0.864	0.75	0.07
Ever experienced forced sex (%)	15.82	14.94	14.36	0.01	0.02
Lower income (%)	60.31	58.39	57.79	0.00	0.04
Ever married (%)	64.26	64.57	64.88	0.09	0.31
Ever intimate partner violence (IPV)	23.4	23.4	23.3	0.08	0.23
Outcomes (at Wave 4)					
Depression (%)	25.68	25.48	24.90	0.21	0.04
Anxiety (%)	9.16	9.16	9.53	0.99	0.05
Suicide ideation (%)	8.2	8.2	7.8	0.99	0.20
Drug Abuse/Dependence (%)	5.48	6.64	6.51	0.00	0.42
Alcohol Abuse/Dependence (%)	14.82	17.77	18.07	0.00	0.20
Cannabis Abuse/Dependence (%)	5.55	5.51	5.34	0.66	0.27
Opioid Abuse/Dependence (%)	10.66	10.64	10.67	0.94	0.86

Tests report significance (*p*-value) of *t*-tests for equal means. “Wave IV Only”, cases with data at Waves I and IV; “Full Sample”, cases with data at Waves I, III and IV (all waves); “Analysis Sample”, cases with complete data for all analysis variables at all waves.

**Table 2 medicina-55-00741-t002:** Unadjusted mental health outcome prevalence, in percent: ever-pregnant women by Wave IV, Add Health Wave I (Population-Weighted).

Mental Health Problems	All (N = 3935)	Not Yet Pregnant (N = 3619)	Ever Pregnant, No Abortion (N = 219)	Unwanted Pregnancy Abortions (UPA) Only (N = 91)	Ever Wanted Pregnancy (WPA) Abortion (N = 6)	Chi-Square
Depressed	30.1	29.3	35.9	42.7	83.6	0.02
Anxiety	13.6	13.5	14.6	19.0	1.2	0.49
Suicide ideation	17.9	18.1	15.3	18.2	0	0.75
Sum of above three affective disorders	0.62 (0.57–0.66)	0.61 (0.57–0.65)	0.66 (0.49–0.83)	0.80 (0.46–1.14)	0.85 (0.53–1.17)	0.00
Illicit Drug Abuse/dependence	1.3	1.3	0.7	0.3	0	0.63
Opioid abuse/dependence	4.1	4.0	5.5	8.0	0	0.44
Cannabis abuse/dependence	14.5	14.1	15.9	26.4	68.6	0.00
Alcohol abuse/dependence	11.8	11.3	16.4	25.6	0	0.02
Sum of above four substance abuse disorders	0.44 (0.39–0.49)	0.42 (0.37–0.47)	0.61 (0.41–0.82)	0.83 (0.52–1.15)	0.85 (0.31–1.39)	0.00
Sum of above seven mental health problems	1.05 (0.98–1.13)	1.03 (0.95–1.10)	1.27 (0.98–1.57)	1.63 (1.12–2.15)	1.70 (0.93–2.48)	0.00

Chi-square tests row independence excluding “All”. At Wave I, the N for Ever Unwanted Pregnancy Birth is 143 and for Only Wanted Pregnancy Births is 50.

**Table 3 medicina-55-00741-t003:** Unadjusted mental health outcome prevalence, in percent: ever-pregnant women by Wave IV, Add Health Wave III (Population-Weighted).

Mental Health Problems	All (N = 3935)	Not Yet Pregnant (N = 1327)	Ever Pregnant, No abortion (N = 1858)	Unwanted Pregnancy Abortions (UPA) Only (N = 624)	Ever Wanted Pregnancy (WPA) Abortion (N = 126)	Chi-Square
Depressed	20.4	17.6	22.0	22.1	17.3	0.07
Anxiety	13.1	10.9	14.2	13.7	19.2	0.06
Suicide ideation	6.7	8.1	5.2	8.8	7.4	0.04
Sum of above three affective disorders	0.40 (0.37–0.44)	0.37 (0.31–0.42)	0.41 (0.37–0.46)	0.45 (0.37–0.52)	0.44 (0.27–0.60)	0.00
Illicit Drug Abuse/dependence	9.9	10.0	6.9	19.3	18.1	0.00
Opioid abuse/dependence	19.6	17.9	17.8	28.5	30.8	0.00
Cannabis abuse/dependence	18.9	18.3	15.7	31.4	23.5	0.00
Alcohol abuse/dependence	17.0	21.3	11.9	24.3	19.4	0.00
Sum of above four substance abuse disorders	0.99 (0.92–1.07)	0.97 (0.87–1.08)	0.87 (0.78–0.97)	1.44 (1.23–1.66)	1.23 (0.91–1.54)	0.00
Sum of above seven mental health problems	1.40 (1.30–1.49)	1.34 (1.21–1.47)	1.29 (1.16–1.41)	1.89 (1.64–2.14)	1.67 (1.27–2.06)	0.00

Chi-square tests row independence excluding “All”. At Wave III, the N for Ever Unwanted Pregnancy Birth is 1,461 and for Only Wanted Pregnancy Births is 633.

**Table 4 medicina-55-00741-t004:** Unadjusted mental health outcome prevalence, in percent: ever-pregnant Women by Wave IV, Add Health Wave IV (Population-Weighted).

Mental Health Problems	All (N = 3935)	Not Yet Pregnant (N = 0)	Ever Pregnant, No Abortion (N = 2918)	Unwanted Pregnancy Abortions (UPA) Only (N = 807)	Ever Wanted Pregnancy Abortion (WPA) (N = 210)	Chi-Square
Depressed	26.7	N/A	25.8	28.2	37.5	0.04
Anxiety	100.4	N/A	10.0	11.9	9.9	0.48
Suicide ideation	7.8	N/A	6.9	11.0	10.1	0.01
Sum of above three affective disorders	0.45 (0.41–0.49)	N/A	0.43 (0.39–0.47)	0.51 (0.43–0.59)	0.57 (0.42–0.73)	0.00
Illicit Drug Abuse/dependence	7.2	N/A	5.3	15.8	6.1	0.00
Opioid abuse/dependence	12.3	N/A	10.1	21.5	11.9	0.00
Cannabis abuse/dependence	5.6	N/A	4.6	9.2	8.3	0.00
Alcohol abuse/dependence	19.7	N/A	16.0	34.3	23.8	0.00
Sum of above four substance abuse disorders	0.56 (0.51–0.62)	N/A	0.47 (0.42–0.52)	0.93 (0.79–1.07)	0.66 (0.46–0.86)	0.00
Sum of above seven mental health problems	1.01 (0.94–1.09)	N/A	0.90 (0.82–0.97)	1.44 (1.26–1.62)	1.24 (0.93–1.55)	0.00

Chi-square tests row independence excluding “All”. “N/A”, Not Applicable. At Wave IV, the N for Ever Unwanted Pregnancy Birth is 1913 and for Only Wanted Pregnancy Births is 1345.

**Table 5 medicina-55-00741-t005:** Relative risk (RR/IRR) (95% CI) of mental health disorders with abortion relative to childbirth for ever-pregnant women by pregnancy intention (wanted versus unwanted), adjusted for covariates and other pregnancy outcomes: Add Health Waves I, III, and IV (*n* = 3935).

Mental Health Problems	Ever Any Abortion	Unwanted Pregnancy Abortions (UPA) Only	Ever Wanted Pregnancy Abortion (WPA)	WPA/UPA
	RR/IRR (95% CI)	RR/IRR (95% CI)	RR/IRR (95% CI)	RR/IRR (95% CI)
Depression	1.63 (1.21–2.21)	1.35 (0.96–1.89)	2.22 (1.32–3.75)	1.65 (0.94–2.87)
Anxiety disorder	1.15 (0.81–1.62)	1.06 (0.74–1.53)	1.72 (0.85–3.47)	1.61 (0.77–3.36)
Suicide ideation	2.38 (1.55–3.66)	1.94 (1.21–3.11)	3.44 (1.53–7.72)	1.77 (0.73–4.29)
**Sum of above three affective problems**	**1.31 *** (1.13–1.53)**	**1.18 ^1^ (1.00–1.40)**	**1.69 *** (1.31–2.18)**	**1.43 * (1.08–1.89)**
Illicit drug abuse/dependence	3.65 (2.37–5.62)	3.15 (2.05–4.83)	2.60 (1.13–5.95)	0.82 (0.36–1.89)
Opioid abuse/dependence	2.28 (1.67–3.11)	2.13 (1.53–2.98)	1.95 (1.04–3.65)	0.91 (0.47–1.77)
Cannabis abuse/dependence	2.66 (1.80–3.93)	2.74 (1.83–4.08)	2.40 (1.13–5.10)	0.88 (0.40–1.92)
Alcohol abuse/dependence	3.56 (2.52–5.03)	3.70 (2.55–5.37)	2.78 (1.58–4.87)	0.75 (0.42–1.35)
**Sum of above four substance abuse problems**	**2.12 *** (1.81–2.49)**	**2.01 *** (1.69–2.38)**	**1.99 *** (1.53–2.58)**	**0.99 (0.75–1.31)**
**Sum of above seven mental health problems**	**1.74 *** (1.55–1.96)**	**1.61 *** (1.42–1.83)**	**1.84 *** (1.52–2.23)**	**1.15 (0.93–1.41)**

Shown are population-weighted and -averaged panel regression estimates, derived from poisson models for summary measures and from logistic models for individual outcomes. The sample size (N) for each column category is time-dynamic by wave and is reported for abortion outcomes in Table 2, Table 3 and Table 4. Numbers in parentheses report the 95% confidence interval. Asterisks report significance for chi-square test. “RR”, risk ratio; “IRR”, incidence risk ratio; “CI”, confidence interval; “Add Health”, National Longitudinal Study of Adolescent to Adult Health; ^1^
*p* < * *p* < 0.05; ** *p* < 0.01; *** *p* < 0.001. All models are adjusted for all other pregnancy outcomes, the lagged dependent variable at prior waves, and for demographic variables: age (within panel), race, parent education, childhood poverty status and region of origin. Covariates fitted: childhood conditions: 1 = childhood physical abuse, 2 = childhood sexual abuse, 3 = childhood verbal abuse; at Wave I (average age 15): 4 = depression, 5 = anxiety, 6 = suicidal ideation, 7 = alcohol abuse, 8 = drug abuse, 9 = nicotine dependence, 10 = cannabis abuse, 11 = conduct problems in school, 12 = neuroticism, 13 = neighborhood integration, 14 = grade point average (gpa); at Wave IV (average age 28): 15 = ever raped, 16 = relationship satisfaction, 17 = educational attainment; time-dynamic: 18 = respondent poverty income; 19 = marital status; 20 = intimate partner violence.

**Table 6 medicina-55-00741-t006:** Risk (OR/IRR) (95% CI) of mental health disorders with abortion and birth by pregnancy intention (wanted versus unwanted), adjusted for covariates and other pregnancy outcomes: Add Health Waves I, III, and IV (*n* = 3935).

Mental Health Problems	Ever Wanted pregnancy Abortion (WPA)	Unwanted Pregnancy Abortions (UPA) Only	Wanted Pregnancy Births Only	Ever Unwanted Pregnancy Birth	Test for Equal ORs	
					Abortion	Birth
					Wanted = Unwanted	Wanted = Unwanted
	OR/IRR (95% CI)	OR/IRR (95% CI)	OR/IRR (95% CI)	OR/IRR (95% CI)	*p*-value	*p*-value
Depression	1.38 (0.87–2.17)	1.09 (0.83–1.42)	0.62 (0.47–0.81)	0.81 (0.63–1.03)	0.349	0.035
Anxiety disorder	1.27 (0.67–2.40)	1.04 (0.78–1.39)	0.74 (0.55–0.99)	0.98 (0.76–1.26)	0.560	0.047
Suicide ideation	1.20 (0.60–2.39)	1.09 (0.77–1.55)	0.35 (0.23–0.52)	0.56 (0.40–0.78)	0.805	0.011
**Sum of above three affective problems**	**1.20 ^1^ (0.97–1.49)**	**1.05 (0.92–1.19)**	**0.71 *** (0.62–0.81)**	**0.88 * (0.78–0.98)**	0.261	0.001
Illicit drug abuse/dependence	0.92 (0.43–1.94)	2.17 (1.59–2.94)	0.35 (0.23–0.55)	0.69 (0.50–0.95)	0.025	0.001
Opioid abuse/dependence	1.08 (0.63–1.84)	1.55 (1.22–1.98)	0.55 (0.41–0.75)	0.73 (0.57–0.93)	0.203	0.070
Cannabis abuse/dependence	1.17 (0.64–2.15)	1.70 (1.27–2.20)	0.49 (0.33–0.73)	0.62 (0.47–0.83)	0.254	0.206
Alcohol abuse/dependence	1.34 (0.82–2.20)	1.68 (1.29–2.20)	0.48 (0.36–0.65)	0.45 (0.35–0.59)	0.401	0.637
**Sum of above four substance abuse problems**	**1.18 (0.96–1.46)**	**1.43 *** (1.26–1.61)**	**0.59 *** (0.51–0.70)**	**0.71 *** (0.63–0.81)**	0.100	0.016
**Sum of above seven mental health problems**	**1.17 ^1^ (0.99–1.37)**	**1.25 *** (1.14–1.38)**	**0.63 *** (0.57–0.70)**	**0.78 *** (0.71–0.85)**	0.427	0.000

Shown are population-weighted and -averaged panel regression estimates, derived from poisson models for number of mental health outcomes and from logistic models for all other outcomes. Numbers in parentheses report the 95% confidence interval. The sample size (N) for each pregnancy outcome category is time-dynamic by wave and is reported for abortion outcomes in Table 2, Table 3 and Table 4. The test reported is chi-square with one degree of freedom. “RR”, risk ratio; “IRR”, incidence risk ratio; “CI”, confidence interval; “Add Health”, National Longitudinal Study of Adolescent to Adult Health; “Test: Abortion, Wanted = Unwanted”, tests whether column 1 is equal to column 2; “Test: Birth, Wanted = Unwanted, tests whether column 3 is equal to column 4; * *p* < 0.05; ** *p* < 0.01; *** *p* < 0.001. All models are adjusted for all other pregnancy outcomes, the lagged dependent variable at prior waves, and for demographic variables and covariates as listed in Table 5.

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
