# Peer review of "Affective and Substance Abuse Disorders Following Abortion by Pregnancy Intention in the United States: A Longitudinal Cohort Study"

_medicina, 2019, doi:10.3390/medicina55110741_

Round 1
Reviewer 1 Report
The authors have improved the manuscript adequately. However, there are still some errors in the new manuscript. For examine, “1.4. Literature review: the abortion and mental health controversy”; line 163: “adequately represent[]”. Please correct them.
Author Response
Reviewer 1 wrote:
The authors have improved the manuscript adequately. However, there are still some errors in the new manuscript. For examine, “1.4. Literature review: the abortion and mental health controversy”; line 163: “adequately represent[]”. Please correct them.
My response:
I have reworded this citation to remove the ellipsis bracket, and checked and corrected other errors in the manuscript.
Reviewer 2 Report
The author of this manuscript described the longitudinal associations between pregnancy intention and abortion exposure and seven major psychological outcomes, including affective disorders and substance use disorders, using data from the National Longitudinal Survey of Adolescent to Adult Health (Add Health). The overall findings are that abortion with wanted pregnancy was associated with higher risk for affective disorders compared to birth and abortion with only with unwanted pregnancy, while the risk for substance use disorders was not higher compared to birth with unwanted pregnancy only. The research question is interesting and the findings are of potential significant impact. However, some methodology issues need to be fully addressed before publication.
The authors have explicitly compared the characteristics of the full sample and the analytic sample in Table 1. However, a full comparison of the baseline characteristics between the entire Add Health sample (n = 6139), the full sample (n = 4523), and the analytic sample (n = 3935) should be conducted to examine potential selection bias. The description of the outcome variables are not clear. It is unknown whether the diagnoses of the psychological outcomes are incident cases since the prior visit, or a life-time prevalence. It is better that the prevalence and new incidence of the psychological disorders be provided in a table. Bivariate analyses results comparing the incidence/prevalence rate of psychological disorders should be provided before the multivariate analyses were conducted. The covariate selection need justification, and how each variable was operationalized and specified in the model should be explicitly described. The description of statistical analysis is not clear. How the model for logistic regression and Poisson regressions were specified need more clarification. The results presented in Table 2 and Table 3 are nor clear. The reference group, as well as the sample sizes for the comparison and reference groups should be labeled. For Table 3, it is not clear which are the reference groups. It seems that each comparison was made between a selected group and the rest of the analytic sample, which is not the proper way for answering the question of interest. When the experience of abortion was associated with increased risk of mental disorders, the experience of childbearing should be protective, which is self-explanatory. There are errors in the headers and grammar issues here and there throughout the entire manuscript. The authors should do a careful language check.
Author Response
Comments and Suggestions for Authors
The author of this manuscript described the longitudinal associations between pregnancy intention and abortion exposure and seven major psychological outcomes, including affective disorders and substance use disorders, using data from the National Longitudinal Survey of Adolescent to Adult Health (Add Health). The overall findings are that abortion with wanted pregnancy was associated with higher risk for affective disorders compared to birth and abortion with only with unwanted pregnancy, while the risk for substance use disorders was not higher compared to birth with unwanted pregnancy only. The research question is interesting and the findings are of potential significant impact. However, some methodology issues need to be fully addressed before publication.
The authors have explicitly compared the characteristics of the full sample and the analytic sample in Table 1. However, a full comparison of the baseline characteristics between the entire Add Health sample (n = 6139), the full sample (n = 4523), and the analytic sample (n = 3935) should be conducted to examine potential selection bias. The description of the outcome variables are not clear. It is unknown whether the diagnoses of the psychological outcomes are incident cases since the prior visit, or a life-time prevalence. It is better that the prevalence and new incidence of the psychological disorders be provided in a table. Bivariate analyses results comparing the incidence/prevalence rate of psychological disorders should be provided before the multivariate analyses were conducted. The covariate selection need justification, and how each variable was operationalized and specified in the model should be explicitly described. The description of statistical analysis is not clear. How the model for logistic regression and Poisson regressions were specified need more clarification. The results presented in Table 2 and Table 3 are nor clear. The reference group, as well as the sample sizes for the comparison and reference groups should be labeled. For Table 3, it is not clear which are the reference groups. It seems that each comparison was made between a selected group and the rest of the analytic sample, which is not the proper way for answering the question of interest. When the experience of abortion was associated with increased risk of mental disorders, the experience of childbearing should be protective, which is self-explanatory. There are errors in the headers and grammar issues here and there throughout the entire manuscript. The authors should do a careful language check.
Point by point response please see attachment.

Round 2
Reviewer 2 Report
The revision of the manuscript is substantially improved.
Line 66: header should be 1.2
In author's response (in cover letter) "Table 4 decomposes the RR’s into their underlying odds ratios (OR), defined as the odds on being in the indicated state compared to not being in the indicated state.” The definition of odds ratio is incorrect. Odds ratio is defined as the odds of A compared to the odds of B. It is not correct to say that an RR can be decomposed into underlying odds ratios. Although I do have a better understanding of Table 4, the author's explanation and interpretation in the manuscript about odds ratio is improper. Once the definition is clarified correctly, I agree the manuscript can be published.
Author Response
You are absolutely correct. I have revised the definition of odds ratio in the manuscript to define it correctly. Particularly at lines 238-241, which now read:
"Table 4 expresses the corresponding odds ratios (OR), that is, the ratio of the odds of experiencing the indicated psychological outcome (for example, depression) for those in the indicated state (for example, wanted pregnancy abortion) compared to the odds of experiencing the outcome for those not in the indicated state."
I appreciate the correction.